# Effect of Silibinin on Human Pancreatic Lipase Inhibition and Gut Microbiota in Healthy Volunteers: A Randomized Controlled Trial

**DOI:** 10.3390/ijms252312853

**Published:** 2024-11-29

**Authors:** Cristina Ponce Martínez, Elena Murcia García, Horacio Pérez Sánchez, Fermín I. Milagro, José I. Riezu-Boj, Bruno Ramos Molina, María Gómez Gallego, Salvador Zamora, Rubén Cañavate Cutillas, Juan José Hernández Morante

**Affiliations:** 1Unidad de Investigación de Trastornos de la Alimentación, Facultad de Enfermería, Universidad Católica de Murcia, Campus de Guadalupe, Avda. de Los Jerónimos, s/n, 30107 Murcia, Spain; cponce@ucam.edu (C.P.M.); emurcia@ucam.edu (E.M.G.); ruben.canavate@murciaeduca.es (R.C.C.); 2Bioinformatics and High Performance Computing Group, Universidad Católica de Murcia, Campus de Guadalupe, Avda. de Los Jerónimos, s/n, 30107 Murcia, Spain; hperez@ucam.edu; 3Obesity, Diabetes and Metabolism Laboratory, Biomedical Research Institute of Murcia (IMIB), 30120 Murcia, Spain; bruno.ramos@imib.es; 4Department of Nutrition, Food Sciences and Physiology, Center for Nutrition Research, Universidad de Navarra, C/Irunlarrea, 1, 31008 Pamplona, Spain; fmilagro@unav.es; 5Navarra Institute for Health Research (IdiSNA), C/Irunlarrea, 3, 31008 Pamplona, Spain; jiriezu@unav.es; 6Centro de Investigación Biomédica en Red Fisiopatología de la Obesidad y Nutrición (CIBERobn), Instituto de Salud Carlos III, 28029 Madrid, Spain; 7Facultad de Ciencias Sociosanitarias, Campus de Lorca, Av. de las Fuerzas Armadas, 0, Lorca, 30800 Murcia, Spain; mgomezg@um.es; 8Departamento de Fisiología, Facultad de Biología, Universidad de Murcia, C/Campus Universitario, 5, 30100 Murcia, Spain; sazana@um.es

**Keywords:** microbiota, obesity, pancreatic lipase, silibinin, thistle extract

## Abstract

Thistle (*Onopordum acanthium*) has been traditionally employed for liver protection. However, we recently identified silibinin, the main bioactive compound of thistle extract, as an in vitro pancreatic lipase inhibitor, which suggested a potential role as an anti-obesity agent. This study aimed to assess, in vivo, the efficacy, safety, and effects of silibinin on human lipase. As a secondary objective, we evaluated potential changes in gut microbiota after silibinin treatment. A randomized trial comparing 150 mg/silibinin, 300 mg/silibinin, and a thistle extract (equivalent to 150 mg/silibinin) with placebo and orlistat/120 mg was conducted. Fecal fat excretion, clinical parameters, and microbiota changes were analyzed. Orlistat showed the highest fecal fat excretion, although thistle extract had similar results (*p* = 0.582). The 150 mg/silibinin group reported the fewest adverse effects. Both silibinin and orlistat reduced plasma triglycerides (*p* = 0.016) and waist circumference (*p* = 0.001). Specific microbiota changes, such as increases in Mycobacteriaceae and Veillonellaceae, were associated with higher fat excretion. Although the present work was conducted in the short term and in people of normal weight, our results suggest that silibinin may be safe and effective for obesity, with minimal adverse effects and no significant changes in microbiota diversity. Further studies are needed to explore its microbiota-related benefits.

## 1. Introduction

Obesity is a chronic and multifactorial disease characterised by excessive adipose tissue accumulation and represents the most pressing public health challenge [1]. This pandemic has far-reaching consequences, both on an individual and social level, affecting not only the physical health but also the mental well-being and overall quality of life of those suffering from this disorder. The high prevalence, estimated in 2021 in 650 million people with obesity, underscores the gravity of the obesity epidemic, affecting not only high-income countries but also emerging economies and even low-income nations [1,2]. Therefore, it is vital to design accurate interventions with broad application and few adverse effects against this disorder.

One of the leading causes of obesity is the intake of high-fat and high-palatable foods, which can disrupt the balance of energy intake and expenditure, ultimately contributing to an increased body weight and the development of obesity [3]. Traditionally, lifestyle modifications, such as dietary changes and increased physical activity, have formed the cornerstone of obesity management. However, the success of lifestyle modification is limited [4]. Whereas drug treatment is often indicated, there are currently few well-tolerated drugs available that have proven long-term efficacy in maintaining body weight loss [4]. Pharmacological approaches for addressing obesity aim to modulate diverse physiological mechanisms associated with appetite regulation, energy expenditure, and lipid metabolism. Currently, glucagon-like peptide 1 receptor agonists (GLP1Ra) stand as the most effective drug-based obesity treatment, given their efficacy in reducing body weight and blood glucose levels in people who are overweight or obese [5]. However, the subcutaneous administration of GLP1Ras restricts their applicability in certain patient populations, and notable severe adverse effects have been described with their use [6]. Therefore, it is imperative to explore alternative approaches to GLP1Ras. In this context, certain synthetic derivatives of products made by Streptomyces toxytricini, like orlistat or cetilistat, have proven to be a suitable alternative. These drugs are selective inhibitors of gastrointestinal lipases, which are involved in triglyceride hydrolysis, and although their effect on weight loss is limited compared to GLP1 analogues, these drugs are usually associated with lower side effects, characterized by gastrointestinal issues, which also have limited their use [7].

Recently, great interest has been posed in the role of microbiota on obesity development. High-fat and high-sugar diets tend to encourage the proliferation of less favourable bacterial species, including those linked to inflammation and metabolic disorders, which could explain the continued expansion of the obesity epidemic, even though calorie intake appears to be declining worldwide [8]. On the other hand, diets rich in fiber, whole grains, and diverse plant-based foods promote the growth of beneficial bacterial groups, such as *Bifidobacteria* and *Lactobacilli*, which are associated with improved metabolic health and immune function [8,9,10].

In a previous report, we described the in silico and in vitro impact of silibinin, a naturally occurring bioactive compound extracted from thistle (*Onopordum acanthium)*, establishing its role as a pancreatic lipase inhibitor [11]. In this previous work, we described that silibinin interacts with the hydrophobic sites of the catalytic centre of pancreatic lipase in a similar way that orlistat does, which suggests that silibinin has the potential as a complement to dietary therapy for the treatment of obesity [11]. Silibinin has been widely used in clinical treatment for liver disorders and exhibited therapeutic potential for non-alcoholic fatty liver disease [12], and some previous works have described some mechanisms by which silibinin could also act against obesity in animal models and in cellular models [13,14], but to our knowledge, no previous report has evaluated the potential of this bioactive compound in the context of people with obesity.

However, the expansion of the application of this bioactive compound in individuals with obesity needs a rigorous validation of its efficacy and safety, including a potential effect on gut microbiota. Therefore, this study aims to elucidate the impact of silibinin on dietary fat metabolism and gut microbiota composition in healthy volunteers, assessing its influence on pancreatic lipase activity and ensuring its safety profile in normal-weight individuals.

## 2. Results

### 2.1. Effect of Silibinin on Clinical Parameters

One of the initial twelve volunteers dropped out of the study during the first intervention; hence, the present data refer to the final eleven volunteers. During the study period, a total of 68 adverse effects (AEs) were observed throughout the five interventions (placebo, 150 mg silibinin, 300 mg silibinin, thistle extract, and orlistat). The intervention with fewer AEs was silibinin 300 mg (*n* = 9), while a higher number of AEs was observed with orlistat (*n* = 23). All participants recorded at least two AEs during the study duration, with two volunteers declaring up to eleven AEs. Importantly, all these AEs were considered to have low or mild intensity. Notably, despite the occurrence of these AEs, this study did not necessitate the discontinuation of any participant owing to the presence of severe effects. The most common AE was constipation, which was suffered by at least one volunteer in the five study phases. On the contrary, melaena or dysphagia were not described by any participant. Overall, all interventions recorded fewer AEs than orlistat. Table 1 shows the presence of AEs in the different interventions.

In this work, we have also examined potential changes in hunger/satiety perceptions to study a possible relationship between changes in appetite because of the treatments. However, we did not observe a statistically significant association between treatments and hunger feelings since, in all treatments, there were individuals showing higher or lower hunger sensations compared to placebo (Appendix A).

The different treatments did not significantly change the biochemical parameters evaluated in the present work (Figure 1), although plasma triglycerides values were slightly higher after 150 mg silibinin and thistle extract interventions compared to placebo. (Figure 1). No other statistically significant differences were identified in the biochemical parameters, and no statistically significant changes from baseline in vital signs were observed during any of the treatments. Specific data are described in Appendix A.

### 2.2. Effect of Silibinin on Anthropometric Parameters

Figure 2 represents the different anthropometric evaluations throughout the different interventions. There were minimum changes in these parameters. In fact, we did not observe any statistically significant difference in body weight, body fat, and fat-free mass. However, waist circumference was already significantly diminished in all the interventions, including silibinin and orlistat, compared to the waist circumference values at baseline or after the placebo intervention.

### 2.3. Effect of Silibinin on Faecal Fat Excretion

As expected, higher faecal fat excretion was observed with the orlistat treatment (15.8 ± 4.9 g/24 h) (Figure 3), although it is important to highlight the huge response variability among the participants. In fact, although the mean fat excretion in the orlistat intervention was 40.1% higher than that in the placebo, the differences did not reach the statistically significant level (*p* = 0.1520). After orlistat, the treatment with higher faecal fat excretion was the thistle extract, with a 28.2% higher fat excretion than the placebo.

### 2.4. Effect of Silibinin on Gut Microbiota

The analysis overview of the community profiling, determined by the analysis of alpha diversity, showed no statistically significant differences between intervention groups when evaluated by Chao1 (F_anova_ = 0.75082, *p* = 0.58906) and Shannon (between groups: Fanova = 0.28727, *p* = 0.91808), probably determined by the high inter-individual variability (Appendix A). Beta diversity (mean of different families between treatments) also showed no significant differences between groups (F_anova_ = 0.45551, R^2^ = 0.039763, *p* = 1).

To compare specific changes between interventions, we initially studied possible differences regarding the relative abundance of bacterial families and genera between baseline and placebo; however, our data showed no significant changes. Consequently, the effect of the different treatment interventions on microbiota was compared with that of the placebo. The influence of silibinin, thistle extract, and orlistat on microbiota was compared with the placebo through a Multiple Linear Regression, with individuals as covariates (Table 2). The lower silibinin dose (150 mg) was associated with a statistically significant decrease in Lachnospiracea incertae sedis, Alkalibacter, and Acholeplasma. Family Acholeplasmataceae was also reduced. The higher silibinin dose (300 mg) also reflected similar changes on Lachnospiracea incertae sedis, but no further changes were observed. The silibinin-enriched thistle extracts also exerted a slight influence on microbiota. Concretely, a statistically significant increase in genera Dorea, Mycobacterium, Nevskia, Pectinatus, Acidaminococcus, and Rikettsia was observed. Interestingly, the Mycobacteriaceae family was also increased. Moreover, the orlistat intervention (120 mg) most evident change was related to the decrease in Haemophilus, although other genera also decreased. In fact, the Pasteurellaceae family also decreased significantly. In contrast, Ethalonigenens, Oxobacter, and Slackia genera increased because of orlistat (Table 2).

A correlation analysis was performed to evaluate the relationship between microbiota composition and the effect of the different interventions. In this regard, Table 3 shows the correlation coefficients between faecal fat excretion and microbiota composition in all the interventions (*n* = 61 samples). At the family level, a higher content of Veillonellaceae and Flavobacteriaceae was associated with an increase in fat excretion. Several genera, like Selenemonas, Peptoniphilus, and Azospirillum, were also associated with higher fat content in patients’ stools, while other genera, like Salinivibrio, Citrobacter, Lishizhenia and Butyricimonas, showed the opposite trend.

## 3. Discussion

The present study was conducted with the aim of evaluating the in vivo effects of silibinin on pancreatic lipase in healthy volunteers by measuring faecal fat content, as well as a potential effect on gut microbiota. The data obtained indicated that the adequate intervention was based on silymarin, a thistle extract enriched in silibinin (similar to a 150 mg dose of silibinin), since the effect on the anthropometric parameters, like the decrease in waist circumference, was similar to orlistat, but the number of adverse effects was much lower.

As reported in previous investigations using silibinin [15], most of the side effects observed were gastrointestinal issues, conceivable attributable to lipase inhibition. Notably, the well-known adverse effects of orlistat, including oily stools, diarrhoea, abdominal pain, and faecal spotting [16], have been associated with a high drop-out ratio of patients using orlistat [17]. Furthermore, a comprehensive meta-analysis has described an elevated likelihood of adverse effects with orlistat compared to other anti-obesity agents [18]. Consequently, although the present study was conducted only in the short-term and in normal-weight people, the relatively lower number of adverse effects noted with silibinin or thistle extract suggest their potential for use in individuals with obesity, though further research are needed to confirm their long-term safety and efficacy.

Previous reports have described the role of orlistat on hunger/satiety perception, potentially explained by the raised postprandial secretion raise of GLP-1 [19,20]. Nevertheless, we did not observe any significant change in hunger/satiety signals, which could be related to the employment of standardized diets or inter-individual variability.

To further evaluate the safety of these interventions, changes in biochemical and anthropometrical parameters were also evaluated. In this line, no intervention induced a significant change in these parameters. Curiously, a slight but significant increase in triglycerides was observed regarding silibinin and thistle extract, but not with orlistat [21]. In fact, during the milk thistle extract intervention, plasma triglyceride values exceeded current recommendations (less than 150 mg/dL). These data may indicate a higher triglyceride clearance rate from the liver since previous reports have described a protective role of silibinin on liver metabolism, associated with a reduction in liver fat content [15]. In this regard, it should be noted that at baseline and after the intervention, plasma triglycerides were within normal ranges. Therefore, although there was a statistically significant change from a physiological perspective, this modification was inappreciable. Waist circumference was also reduced in all interventions, which may indicate that the inhibition of pancreatic lipase produced a decrease in fat accumulation at the abdominal level, as observed in previous studies with orlistat [22,23]. However, it is important to note that orlistat does not specifically target fat in the abdominal region. The reduction in waist circumference is a consequence of the overall decrease in body fat resulting from the lower absorption of dietary fats.

The main outcome of the present study was faecal fat excretion as a measurement of pancreatic lipase inhibition. As expected, a higher effect was observed with orlistat as a positive control; however, the thistle extract was the next intervention with a higher effect on lipase inhibition. Nevertheless, there were no statistically significant differences in any intervention, probably associated with the controlled diet and the baseline physiological status of participants. Several mechanisms have been proposed to explain the effect of silibinin on pancreatic lipase. Previously, we were able to demonstrate the inhibition of pancreatic lipase by silibinin in silico and in vitro. The present data are in line with the in vitro observations [11]. Other authors have also described other effects of silibinin that may rely on this observation. For instance, Kheong et al. carried out a randomized trial, and they found that silymarin, whose main component is silibinin, may have the potential to influence lipid (fat) metabolism [24]. It may affect the absorption of dietary fats and regulate lipid levels in the blood. It appears to be safe and well tolerated. Considering that the effect of thistle extract on fat excretion was not inferior to that caused by orlistat and the lower number of adverse effects, in our opinion, the employment of this bioactive compound would be more suitable for patients with obesity.

Considering that the silibinin content in the silymarin thistle extract was similar to the lower silibinin dose (150 mg), but the net effect on fat excretion was much higher on the extract, we hypothesize that other functional compounds of the thistle may also be acting on pancreatic lipase [25]. Previous reports have described that thistle milk contains many different flavonolignans, like silybin A and B, isosilybin A and B, silychristin, and silydianin, among others [26]. While silibinin stands as the main active compound of thistle extract, emerging evidence suggests that other components, like isosylibin, contribute to lipid metabolism through the AMPK/SREBP-1c/PPARα pathway [27]. Moreover, Awla et al. have shown that rats treated with a thistle extract exhibited preventive effects, including decreased glucose levels, attenuated weight gain, and reduced blood pressure [28]. Nevertheless, further studies will be necessary to demonstrate the effect of these compounds on human pancreatic lipase.

A secondary aim of the present work was to evaluate the effect of silibinin on gut microbiota since, on the one hand, drastic changes may limit the use of this compound, and on the other hand, the effectiveness of silibinin could be mediated by these changes. Note that there were no significant changes in global diversity in any intervention. Partly, this observation is a consequence of the high inter-individual variability. However, although there were no global changes, several taxa changed because of the interventions. It is interesting to note that *Lachnospiracea incertae sedis* genus was modified with the two silibinin interventions, considering that Lachnospiraceae are involved in the production of SCFAs (butyrate, propionate, and acetate), which are an important energy source for intestinal epithelial cells, the results may indicate a beneficial effect of silibinin regarding microbiota composition [29,30,31,32].

The thistle extract, on its part, was also able to increase the content of *Dorea*, *Mycobacterium*, *Nevskia*, *Pectinatus*, *Acidaminococcus*, and *Rikettsia* genera. Interestingly, the Mycobacteriaceae family was also increased. The physiological role of these changes is still unknown. For instance, while several species within Mycocobacteriaceae have been associated with higher interferon-mediated inflammation [33], there are also non-pathogenic species, like *Mycobacterium vaccae*, that have been explored for their potential health benefits, including antidepressant-like effects and [34] immune system enhancer [35]. Interestingly, *Acidaminococcus* has also been associated with the anti-obesity effects of green tea polyphenols [36], which reinforces the benefits of silibinin on patients suffering from obesity.

Orlistat also induced several changes. Pasteurellaceae and *Haemophilus* were reduced, while *Oxobacter*, *Slackia*, and *Ethalonigenens* were increased after orlistat administration. Again, the net physiological effects of these changes are unknown since bacteria like Pasteurellaceae have been described as inflammatory markers [9], while other genera are involved in phytoestrogen metabolism and have been described as beneficial for menopause and other estrogen-related pathologies [37].

The only previous study evaluating the effect of orlistat on microbiota was conducted by Uehira et al., but this study only evaluated long-term changes (3 months of orlistat 60 mg t.i.d.). They describe several changes, especially regarding *Lactobacillus* and Firmicutes, but one important observation of this previous work was that several microbiota components were responsible, at least in part, for the effect of this drug [7]. Therefore, we also evaluated the relationship between the microbiota and the net effect of the interventions, measured as faecal fat content. In this regard, the amount of Veilloneaceae and Flavobacteria was positively correlated with faecal fat content, which suggests that the effect of silibinin on pancreatic lipase could be more pronounced in individuals with a higher proportion of these bacteria in their gut microbiota. In contrast, *Salinivibrio*, *Citrobacter*, *Lishizhenia*, and *Butyricimonas* were associated with lower fat excretion. To the best of our knowledge, this is the first report showing the relation between microbiota composition and fat loss; however, some reports described similar observations that reinforce our observations. In this regard, Palmas et al. described that the less abundant taxa in obese patients was Flavobacteriaceae, and its abundance was negatively correlated with fat mass, waist circumference, and BMI [38]. In the same line, Palmas et al. have observed an increase in Veilloneaceae abundance after weight loss, which as a whole suggests that the relative abundance of these species will be essential for adequate lipid metabolism, or at least, be relevant to increasing the effectiveness of weight loss interventions [39]. Strategies aimed to enhance their abundance could offer therapeutic potential in overweight patients by modulating lipid digestion and possibly preventing further weight gain.

At this point, several drawbacks of the present study should be commented on. It is important to note the limitations of our study, such as the small sample size; however, it is similar to the sample size of previous studies [7]. The duration of the intervention was intended to be short term, as it has also been conducted in previous studies [3]. Therefore, the data should be interpreted in this context. Longer duration studies could provide other information; however, the objective of this study was to confirm that silibinin did not significantly alter the gut microbiota composition, which can be derived from the present observations. We also aimed to control the highest number of confounding parameters possible, including dietary characteristics; therefore, the designed intervention was considered the most appropriate. Nevertheless, various studies have shown that even short periods are enough to induce gut microbiota changes [7,40,41]. Another potential limitation is the fact that the participants were healthy individuals (i.e., not overweight or obese) whose diet, serum biochemistry, and physiological parameters, in general, were within normal values. Since orlistat and similar pharmacological interventions are intended for people requiring weight loss, this could be considered a limitation of this study.

Finally, although several relationships were observed regarding microbiota composition and fat excretion, we cannot conclude that the effect of silibinin on pancreatic lipase was due to changes in microbiota, but these observations pave the way for further research aimed at determining how the microbiota is able to regulate the in vivo activity of certain bioactive compounds. This insight could significantly contribute to the understanding of the complex interactions between natural compounds and gut microbiota, offering potential advancements in therapeutic strategies within the field of phytomedicine. Overall, while the present results are promising, further research is needed to confirm the efficacy of silibinin over time.

## 4. Materials and Methods

### 4.1. Study Design

This study was a double-blind, randomized, placebo-controlled, crossover pilot trial conducted on healthy men and women volunteers. Individuals were randomized to receive a placebo, one of a range of silibinin doses, a thistle extract (silymarin), or orlistat for 3 days, where participants were maintained on a precisely controlled diet along the different interventions. Each subject received three standardized meals daily with 30% of calories from fat (see Appendix A). On the intervention days, meals were distributed at breakfast, lunch, and dinner, along with one of the indicated doses according to their randomization. Meals were distributed at 08:30, 13:30, and 20:00, trying to adjust to the patients’ habits. A nurse was present during the distribution of meals to ensure the intake of the compound.

All treatments were delivered on green capsules, size 0, made with hard gelatin (head and body) (provided by Guinama Ltd., Valencia, Spain). These capsules are capable of disintegrating in less than 15 min and meet the requirements of the European Pharmacopoeia and USP. Capsucel (microcrystalline cellulose, Guinama Ltd., Valencia, Spain) was employed as an excipient in all treatments. Silibinin and orlistat were purchased from Merck (Merck KGaA, Darmstadt, Germany). Silymarin, a thistle extract with 50% silibinin content, was obtained from Metapharmaceuticals (Metapharmaceuticals Ltd., Barcelona, Spain). The treatments were made by two researchers, who were not present at the time of administration. These researchers also carried out the randomization, according to a Latin Square procedure [42], to ensure that all participants followed all the interventions. Randomization was performed by a member of the research group (J.J.H.M.) with Excel software (version 365, Microsoft Co. Redmond, WA, USA), with a visual basic Macro developed to this end, as previously described [43]. The protocol of this randomized trial adheres to the CONSORT guidelines [44]. The supporting CONSORT checklist for this trial is available as Appendix A. The trial (#NCT05069298) was registered at: https://clinicaltrials.gov/ (accessed on 25 November 2024).

Each study comprised an initial screening visit one week prior to the start of this study, with a 1-day run-in period, a 3-day treatment period, and a 1-day post-treatment follow-up visit. The initial evaluation was carried out by a physician, where a complete physical examination was performed. This allowed us to exclude the presence of any disease or disorder in the participants; hence, all volunteers took part in the study. Figure 4 represents the flow diagram of the present work.

### 4.2. Outcome Measurement

The main endpoint of the effectiveness of this study was the daily fat excretion in faeces. Secondary endpoints were gut microbiota variability, clinical signs, tolerability (gastrointestinal adverse effects), and clinical laboratory parameters, in particular, cholesterol, high-density lipoprotein (HDL-C), low-density lipoprotein (LDL-C), and triglycerides.

### 4.3. Participants

Men and women volunteers aged 18–45 years, with a body mass index lower than 25 kg/m^2^ (within normal weight range), no clinically relevant abnormalities, and no relevant systemic disease history, were considered for inclusion in these studies. Exclusion criteria included the presence of any clinically relevant symptoms or severe disease within 4 weeks of the start of the study, any history of hepatic dysfunction, and any condition that might affect the pharmacokinetics properties of the studied drugs. In addition, diarrhoea (>2 liquid stools per day) or constipation (3 days duration) 1 week prior to the start of the study, history of hypersensitivity to lipase inhibitors, evidence of hepatitis B or C, HIV positivity, smoking (>4 cigarettes/day), history of alcohol or substance abuse, bulimia or laxative abuse, were also considered as exclusion criteria. The use of antibiotics or other drugs that may alter gut microbiota during the study period was also considered as exclusion criteria.

A total of 12 individuals were randomized (7 women). All volunteers received the same weekly treatment, alternating between a placebo, 150 mg of silibinin, 300 mg of silibinin, 300 mg of thistle extract (equivalent to 150 mg of silibinin), and 120 mg of orlistat. One subject withdrew from the study before ending the first intervention, and therefore, it was excluded from the final sample. Participants received a grant of 50 € for their participation in the study.

This study was carried out after receiving written authorization from the Ethics Committee of the Catholic University of Murcia (code: CE072111). Participants were informed, either orally or in writing, about the study design. They also explained the ethical aspects of the project, informing the patients about the main objective of the study and guaranteeing the confidentiality and anonymity of the data in accordance with the Declaration of Helsinki and Biomedical Research Spanish Law. All participants provided written informed consent.

### 4.4. Anthropometric and Clinical Measurements

Anthropometric measurements were conducted using established and validated techniques, according to the SEEDO Guidelines [45]. The height was measured with a TANITA rod (model Harpender), with the subject barefoot, erect, and with the head aligned according to the Frankfurt plane. Body composition, including weight, body fat, total body water, and fat-free mass, was evaluated through bioimpedance in accordance with the manufacturer’s instructions (TANITA, MC-780MA). The BMI was also estimated. Blood pressure readings, both systolic and diastolic, were obtained using conventional methods with a digital sphygmomanometer, adhering to the criteria set by the World Health Organization and the International Society of Hypertension. A physician (M.G.G.) collected data related to lifestyle factors, smoking status, and medical history. Participants’ appetite was assessed using a self-reported question specifically designed to compare subjective perceptions of hunger in response to the different interventions. After each intervention, participants were asked to respond to the following question: “Regarding the previous intervention, how do you rate your appetite?”. Responses were recorded using an ordinal scale with three categories: “Less appetite than the previous intervention”, “Same as the previous intervention”, and “More appetite than the previous intervention”. Clear instructions were provided to participants to ensure consistency in responses and minimize potential interpretive bias. Responses were coded and analysed to identify patterns of appetite change associated with the different interventions.

### 4.5. Biochemical Parameters

Blood samples were collected after 12 h of overnight fasting. Glucose, total cholesterol, low-density lipoprotein cholesterol (cLDL), high-density lipoprotein cholesterol (cHDL), triglycerides, alanine-aminotransferase (ALT), and aspartate-aminotransferase (AST) were tested in an automatized Pointcare M4 analyzer (MNCHIP Technologies Co., Ltd., Tianjin, China) by using suitable kits provided by the company (MNCHIP Technologies Co., Ltd., Tianjin, China).

### 4.6. Faecal Sample Collection and Metagenomic Data

The phase of each treatment lasts for 3 days with a post-treatment follow-up period of 1 day. The primary outcome measure is based on daily faecal fat excretion. For the data analysis, different faecal fat samples are collected as follows: at baseline (1–2 days before the first intervention), a faecal sample was obtained to have a reference value. Herein, and then on the subsequent days (days 1, 2, and 3), faecal fat content was determined. In the remaining phases, samples were collected on days 1, 2, and 3.

For microbiota analysis, fecal samples were obtained at baseline (before the first intervention) and one day after every intervention (day 4). Fecal samples were self-collected by the volunteers using OMNIgene.GUT kits from DNA Genotek (Ottawa, ON, Canada) and following the standard guidelines from the supplier. Samples were aliquoted and stored at −80 °C. The isolation of DNA was achieved with the QIAamp^®^ DNA kit (Qiagen, Hilden, Germany) following the manufacturer’s protocol. Bacterial DNA sequencing was carried out at CIMA LAB Diagnostics (University of Navarra, Spain). dsDNA was characterized by Qubit (Thermo Fisher Scientific, Paisley, UK). The Illumina 16S protocol based on the amplification of the V3 and V4 variable regions of the 16S rRNA gene was followed for sequencing. This process consists of two PCRs. In the first one, 12.5 ng of genomic DNA and the 16S Amplicon PCR Forward and 16S Amplicon PCR Reverse primers were used (from Nextera^®^ XT DNA Index Kit FC-131-1002 Illumina (San Diego, CA, USA). The protocol in this first PCR was 95 °C for 3 min and 25 cycles of 95 °C for 30 s, 55 °C for 30 s, 72 °C for 30 s, then, finally, 72 °C for 5 min and hold at 4 °C. The protocol for the second PCR was 95 °C for 3 min, eight cycles of 95 °C for 30 s, 55 °C for 30 s, 72 °C for 30 s, then, finally, 72 °C for 5 min and hold at 4 °C. The PCR quality was assessed in a Labchip Bioanalyzer (Agilent Technologies Inc., Santa Clara, CA, USA). Once the sequencing of all the samples was achieved, up to 40 samples were multiplexed in each run of 2 × 300 cycles. Equimolar concentrations of each sample were mixed, and the pool was diluted up to 20 pM. A total of three runs were performed on the MiSeq sequencer using the MiSeq^®^ Reagent Kit v3 (San Diego, CA, USA) (600 cycles) MS-102-3003. During the process, negative controls were included. To avoid the batch effect, samples were randomized by sex, age, and treatment. Sequence pre-processing was performed using Illumina BaseSpace Sequence Hub software (version 7.28.0. Illumina, San Diego, CA, USA). The taxonomic assignment of the ASV abundance matrix sequences was carried out using the RDP database. The sequencing data of this study can be found in the NCBI SRA repository (accession number PRJNA623853). Alpha diversity was determined by Shanon and Chao1 methods. Beta diversity (mean of different families between treatments) was calculated using the Bray–Curtis index and PERMANOVA test.

The analysis of the differences in abundance between the groups at the taxonomic level of genus and family was carried out by multiple linear regressions using the MicrobiomeAnalyst tool (https://www.microbiomeanalyst.ca/, accessed on 25 November 2024). Raw counts normalized by Centered Log Ratio (CLR) were used.

### 4.7. Statistical Analysis

Population size was calculated using the software GPower 3.1 (Düsseldorf, Germany) [46]. A priori power analysis of F-tests was performed to control for Type 1 and Type 2 probability errors. The sample size was estimated based on the variance observed in previous work [3]. The estimated minimum sample size was nine subjects. Our sample size yielded a power greater than 80%, allowing the detection of true within-group differences with a partial effect size η^2^ ≥ 0.5. To evaluate differences between treatment groups in primary and secondary efficacy measures, a repeated measures analysis of variance (ANOVA) was utilized. The false discovery rate (FDR) method was employed post hoc to account for multiple tests in the primary pharmacodynamic variable. All analyses were performed with the IBM SPSS 27.0 Software (SPSS Inc., Chicago, IL, USA) and Graphpad Prism 8.0 (Graphpad Software, Boston, MA). The significance level was established at *p* < 0.050.

## 5. Conclusions

In conclusion, although the present study was conducted only in the short-term and in normal-weight people, the results of this study suggest that silibinin, a natural compound extracted from thistle (*Onopordum acanthium*), exhibits promising benefits in weight loss. The significant reduction in waist circumference indicates a positive impact on body fat redistribution and, consequently, on obesity-related comorbidities. Furthermore, the lower number of adverse effects in subjects treated with silibinin (both at 150 mg and 300 mg doses) compared to orlistat is a significant advantage, highlighting its potential as a safe and well-tolerated option for weight loss, although this observation needs to be confirmed at long-term. These results are consistent with research suggesting that natural compounds, such as silibinin, may offer safer and better-tolerated alternatives compared to some conventional pharmacological agents. A particularly noteworthy aspect of our study is the evaluation of gut microbiota. Unlike some treatments that have demonstrated alterations in microbial composition, silibinin showed no adverse effects on participants’ gut microbiota. This finding is relevant in the current context of research on metabolic health. Preserving the diversity and stability of the microbiota with silibinin supports its profile as a safe compound for long-term use. In this line, we described that specific microbiota changes, such as increases in Mycobacteriaceae and Veillonellaceae, were associated with higher fat excretion. In summary, the results obtained in this study support silibinin as a promising compound for weight loss, highlighting its efficacy, safety, and potential beneficial impact on intestinal microbiota. These findings may have significant implications for the development of new therapeutic strategies for obesity, providing a solid foundation for future research and clinical applications. Further long-term studies are nonetheless required in order to confirm the effectiveness of silibinin in the context of weight loss.

## Figures and Tables

**Figure 1 ijms-25-12853-f001:**
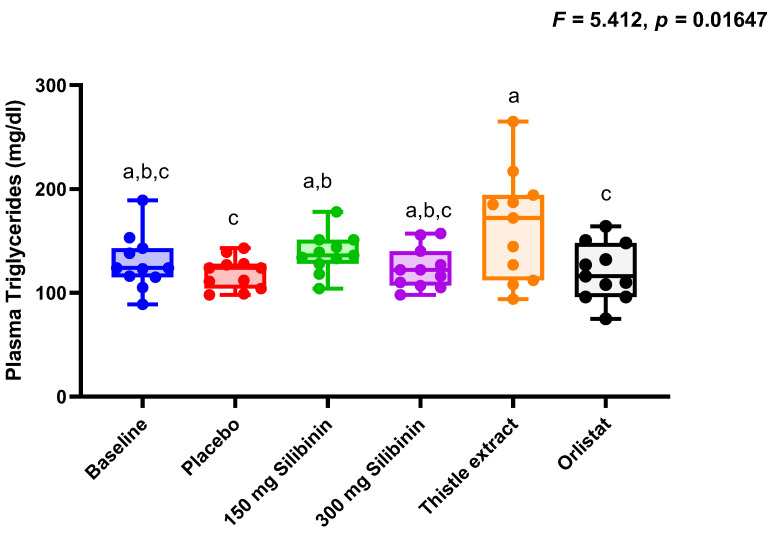
Boxplots showing individual values of the plasma triglycerides throughout the different interventions. Statistical differences were evaluated by a one-way ANOVA. FDR was conducted to control for multiple comparisons. Significance level was *p* < 0.050, after adjustment, in all cases. Boxplots with different letters indicate statistically significant differences after adjustment.

**Figure 2 ijms-25-12853-f002:**
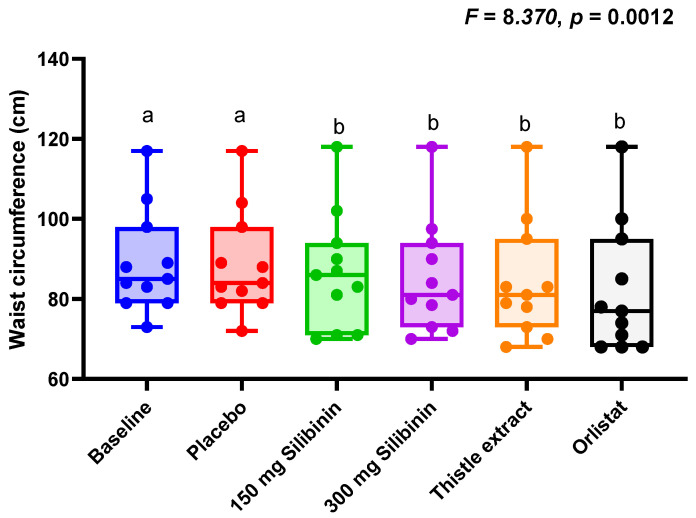
Boxplots showing individual values of waist circumference throughout the different interventions. Statistical differences were evaluated by a one-way ANOVA. FDR was conducted to control for multiple comparisons. Significance level was *p* < 0.050, after adjustment, in all cases Boxplots with different letters indicate statistically significant differences after adjustment.

**Figure 3 ijms-25-12853-f003:**
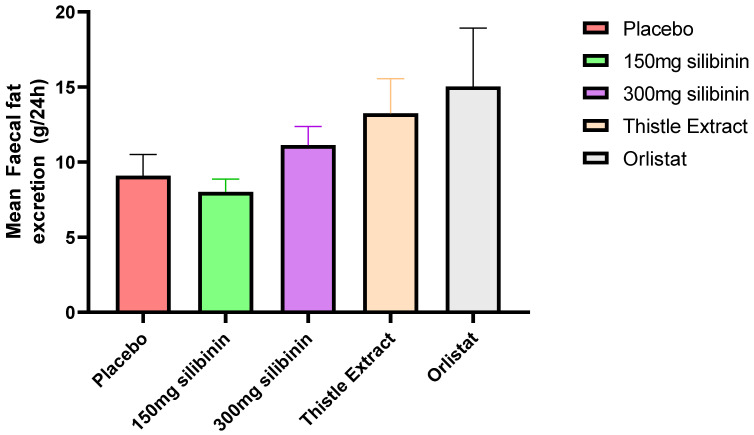
Faecal fat excretion (mean ± sd of the 3 days of treatment) measurements of the different interventions carried out in the present study. Faecal fat excretion was estimated daily, and faecal fat was estimated as the mean fat excretion of the three days. Therefore, a nested one-way ANOVA was conducted. FDR was calculated to control for multiple comparisons; however, there were no statistically significant differences among treatments.

**Figure 4 ijms-25-12853-f004:**
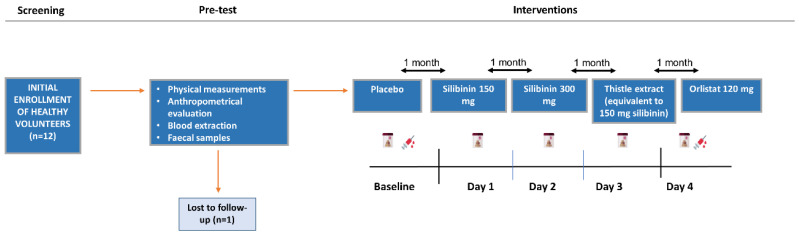
Flow diagram of the present study.

**Table 1 ijms-25-12853-t001:** Comparison of side effects among placebo, 150 mg silibinin, 300 mg silibinin, thistle extract, and orlistat treatments.

	Placebo	150 mg Silibinin	300 mg Silibinin	Thistle Extract	Orlistat	Total
Abdominal pain	1	1	1	3	2	8
Change in bowel habits	1	4	0	2	4	11
Constipation	4	3	4	1	3	15
Diarrhoea	0	0	0	1	3	4
Dysphagia	0	0	0	0	0	0
Dry mouth	1	4	2	2	2	11
Flatulence	2	0	1	1	4	8
Melaena	0	0	0	0	0	0
Vomiting/nausea	0	1	0	0	2	3
Headache	0	1	1	1	0	3
Fatty stools	0	0	0	0	3	3
Other	1	1	0	0	0	2
Total	10	15	9	11	23	68

The data represent the number of side effects in the study.

**Table 2 ijms-25-12853-t002:** Bacterial taxa were significantly different between placebo and the different interventions, analysed by multiple linear regression, using individuals as the covariate.

Bacteria Name	Log2CF	*p*-Value	FDR
Placebo vs. silibinin (150 mg)			
Genus			
*Lachnospiracea incertae sedis*	−0.897	0.00237	0.0138
*Alkalibacter*	−1.54	0.00862	0.0415
*Acholeplasma*	−1.51	0.00902	0.0430
Family			
Acholeplasmataceae	−1.51	0.00902	0.0431
Placebo vs. silibinin (300 mg)			
Genus			
*Lachnospiracea incertae sedis*	−0.788	0.00907	0.0434
Placebo vs. thistle extract			
Genus			
*Dorea*	1.45	0.00254	0.0146
*Mycobacterium*	1.41	0.00489	0.0257
*Nevskia*	0.961	0.00540	0.0277
*Pectinatus*	0.699	0.00922	0.0435
*Acidaminococcus*	0.736	0.01040	0.0480
*Rikettsia*	1.15	0.01090	0.0499
Family			
Mycobacteriaceae	1.41	0.00489	0.0254
Placebo vs. orlistat (120 mg)			
Genus			
*Propionispora*	−1.36	0.00367	0.0203
*Haemophilus*	−3	0.00382	0.0210
*Desulfosporomusa*	−1.33	0.00409	0.0222
*Ethanoligenens*	1.22	0.00592	0.0300
*Dysgonomonas*	−0.565	0.00682	0.0337
*Oxobacter*	0.887	0.00750	0.0367
*Slackia*	1.31	0.00850	0.0408
*Veillonella*	−1.95	0.00990	0.0464
Family			
Pasteurellaceae	−2.66	0.00759	0.0373

Log2CF: logarithm (with base 2) of the fold change. *p*-Value: significance level associated with the multiple linear regression analysis. FDR: False discovery rate.

**Table 3 ijms-25-12853-t003:** Bacterial taxa significantly correlated with fecal fat excretion among the different interventions. Data represent Spearman’s correlation coefficients.

Bacteria Name	Coefficient	Sig
Family		
Veillonellaceae	0.259	0.0435
Flavobacteriaceae	0.259	0.0439
Genus		
*Azospirillum*	0.359	0.0046
*Lachnoanaerobaculum*	−0.318	0.0124
*Butyricimonas*	−0.307	0.0161
*Lishizhenia*	−0.300	0.0187
*Peptoniphilus*	0.296	0.0207
*Citrobacter*	−0.280	0.0290
*Selenomonas*	0.264	0.0397
*Salinivibrio*	−0.260	0.0428
Species		
*Massiliomicrobiota timonensis*	−0.332	0.0090
*Halothermothrix orenii*	−0.295	0.0210
*Clostridium XVIII*	−0.286	0.0254
*Clostridium ramosum*	−0.279	0.0292
*Clostridium populeti*	0.274	0.0326
*Clostridium scindens*	0.265	0.0387
*Marvinbryantia formatexigens*	−0.263	0.0402
*Dysgonomona salginatilytica*	−0.262	0.0413
*Alistipes* sp.	−0.255	0.0473
*Peptoniphilus grossensis*	0.252	0.0498

## Data Availability

All data are available upon a reasonable request to Juan José Hernández Morante (jjhernandez@ucan.edu).

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
