# Peer review of "Effect of Silibinin on Human Pancreatic Lipase Inhibition and Gut Microbiota in Healthy Volunteers: A Randomized Controlled Trial"

_ijms, 2024, doi:10.3390/ijms252312853_

Round 1
Reviewer 1 Report
Comments and Suggestions for Authors
Manuscript titled “Effect of silibinin on human pancreatic lipase inhibition and gut microbiota in healthy volunteers: a randomized controlled trial” reports a study on human participants, who were treated with either orlistat, a thistle extract or silibinin, in order to analyze the effects on weight loss and intestinal microbiota of the latter two treatments. The work is interesting and relevant and the manuscript is properly written; there are some comments and suggestions for the authors:
1. In Figures 1 and 2, the color of the blue, red and purple boxes makes it impossible to see the individual data points within them, please consider changing it to a different tone, similar to how the green, orange and black boxes are presented, where the inner color does not obstruct the view.
2. Related to the previous comment, in Figure 3, please consider maintaining the same colors for each treatment, as compared to the previous ones. For example, Figures 1 and 2 show placebo in red, while it changes to blue in Figure 3. (Also maintain consistency with supplementary material).
3. Lines 108-112 (and supplementary material) mention that hunger/satiety was analyzed. How exactly was the participants’ hunger/satiety analyzed? Please describe this in materials and methods, since this is not specifically stated.
4. In line 151, please add units to the value shown for orlistat (15.8 ± 25.2), moreover, is this value correct? According to Figure 3, it appears that the error should be approximately 2.5 - 3.0 instead of 25. Please check this result.
5. In Table 3, please indicate the meaning of the asterisks.
6. Line 216 highlights the suitability of long-term use of the treatments in obese individuals. Although potentially true, the findings reported in the present study were obtained in shorter periods and from normo-weight individuals, thus, this phrase is not entirely supported by the data, please consider rephrasing.
7. Line 226 mentions previous evidence regarding liver lipids. Your findings (an increase in triglycerides) may not necessarily contradict those mentioned from the literature (hepatoprotection), since changes on the liver and a protective effect on this organ may still be occurring, even if circulating triglyceride concentration increases.
8. Lines 305-324 acknowledge some limitations of the study. Another potential limitation not discussed, is the fact that the participants were healthy individuals (i.e., not overweight or obese), whose diet, serum biochemistry and physiological parameters in general appear to be within normal values. Since orlistat and similar pharmacological interventions are intended for people requiring weight loss, this could be considered a limitation of the study.
9. Similar to comment 6, line 467 of the conclusion mentions long-term suitability, which is not specifically supported by the data of the present work. Please consider rephrasing.
Author Response
COMMENTS TO REVIEWER 1.
(The reviewer’s comments are highlighted in bold characters, and the replies are written in italic characters).
Comments and Suggestions for Authors
Manuscript titled “Effect of silibinin on human pancreatic lipase inhibition and gut microbiota in healthy volunteers: a randomized controlled trial” reports a study on human participants, who were treated with either orlistat, a thistle extract or silibinin, in order to analyze the effects on weight loss and intestinal microbiota of the latter two treatments. The work is interesting and relevant and the manuscript is properly written; there are some comments and suggestions for the authors:
- In Figures 1 and 2, the color of the blue, red and purple boxes makes it impossible to see the individual data points within them, please consider changing it to a different tone, similar to how the green, orange and black boxes are presented, where the inner color does not obstruct the view.
We appreciate the comment of the reviewer. It seems that there was an error with the graphs when they were copied into the document, which we were not able to notice. We have modified these graphs according to your instructions.
- Related to the previous comment, in Figure 3, please consider maintaining the same colors for each treatment, as compared to the previous ones. For example, Figures 1 and 2 show placebo in red, while it changes to blue in Figure 3. (Also maintain consistency with supplementary material).
It is a good remark. Therefore, we have changed figure 3 and supplementary Figure 1 colors to make the figures consistent. However, we are not able to modify the supplementary figure 2 colors, since this figure was generated with microbiome analyst software, and to the best of our knowledge, it is not possible to change such colors.
- Lines 108-112 (and supplementary material) mention that hunger/satiety was analyzed. How exactly was the participants’ hunger/satiety analyzed? Please describe this in materials and methods, since this is not specifically stated.
We regret the lack of information on this regard on the original version of the paper. We have included this information in the Section 4.4 Anthropometric and clinical measurements of the revised paper.
- In line 151, please add units to the value shown for orlistat (15.8 ± 25.2), moreover, is this value correct? According to Figure 3, it appears that the error should be approximately 2.5 - 3.0 instead of 25. Please check this result.
Again, the comment of the reviewer is very accurate. We have included g/24h, which are the units of fecal fat excretion. Moreover, we have corrected the mistake on the standard deviation data of orlistat, which was 4.9.
- In Table 3, please indicate the meaning of the asterisks.
We appreciate the comment. In Table 3, we included only the correlations with statistical significance, which in the program are represented with an asterisk. It seems that when copying them, we forgot to remove the asterisk. Therefore, we have deleted the asterisk in the revised paper.
- Line 216 highlights the suitability of long-term use of the treatments in obese individuals. Although potentially true, the findings reported in the present study were obtained in shorter periods and from normo-weight individuals, thus, this phrase is not entirely supported by the data, please consider rephrasing.
This is a good remark. We have modified this sentence to further clarify this issue as follows:
“Consequently, although the present study was conducted only at the short-term and in normal weight people, the relative lower number of adverse effects noted with silibinin or thistle extract suggest their potential for use in individuals with obesity, though further researches are needed to confirm their long-term safety and efficacy.”
- Line 226 mentions previous evidence regarding liver lipids. Your findings (an increase in triglycerides) may not necessarily contradict those mentioned from the literature (hepatoprotection), since changes on the liver and a protective effect on this organ may still be occurring, even if circulating triglyceride concentration increases.
Effectively, this sentence may lead to confusion. Therefore, we have modified to try to explain our result.
- Lines 305-324 acknowledge some limitations of the study. Another potential limitation not discussed, is the fact that the participants were healthy individuals (i.e., not overweight or obese), whose diet, serum biochemistry and physiological parameters in general appear to be within normal values. Since orlistat and similar pharmacological interventions are intended for people requiring weight loss, this could be considered a limitation of the study.
Indeed, we forgot to include this as a limitation. In fact, with the reviewer's permission, we have taken the liberty of paraphrasing her/him to include this information as a limitation in the reviewed article.
- Similar to comment 6, line 467 of the conclusion mentions long-term suitability, which is not specifically supported by the data of the present work. Please consider rephrasing.
We fully agree with the reviewer. We have rephrased the conclusion accordingly. Changes are highlighted in yellow for a better understanding of the specific modifications.

Reviewer 2 Report
Comments and Suggestions for Authors
Thank you for the opportunity to review this well-written paper. This research provides interesting and valuable insights into silibinin's effects on fecal fat excretion, clinical parameters, and microbiota composition.

The quality of English in the manuscript is generally good, with the main ideas clearly conveyed.
Author Response
COMMENTS TO REVIEWER 2.
(The reviewer’s comments are highlighted in bold characters, and the replies are written in italic characters).
General comments:
Thank you for the opportunity to review this well-written paper. This research provides interesting and valuable insights into silibinin's effects on fecal fat excretion, clinical parameters, and microbiota composition.
Abstract:
- Line 30: Is it silibinina or silibinin?
We regret this mistake. Effectively, the right word is silibinin.
- Lines 37-38: Results cannot be overgeneralized because the study population does not include individuals with obesity. Consider a statement that emphasizes that silibinin was found to be safe and demonstrated effects that may be beneficial for obesity, though the study was conducted in healthy normal-weight individuals.
Indeed, we agree with the reviewer. In fact, we have modified the abstract, as well as the full text, to highlight that the study was conducted in the short term and in healthy people, so long-term studies would be needed to confirm our observations.
Introduction:
- Lines 79-81: The introduction provides a good background, but it could briefly discuss previous in vivo studies on silibinin or similar compounds to establish the study’s novelty.
It is a good remark. We have included three new references [1–3], to better explain the study’s novelty.
- Lines 87-89: Why were normal-weight individuals chosen instead of participants with obesity, given the focus on obesity management? Please explain.
This study was intended to be similar to a Phase I study of a clinical trial. In this phase, drugs, or in this case, silibinin, are delivered to normal weight to try to exclude potential metabolic derangements that may interfere in the results. Nevertheless, further studies will be necessary to confirm our observations, and consequently, this information has been included in the revised paper.
Methods:
- Line 93: Could you provide more justification for the small sample size (n=11) and discuss how this may have influenced the statistical power of the study?
The calculation of the sample size was carried out with the GPower 3.1 software and based on the data of the previous work of Bryson et al [4]. In fact, in this previous work, groups have a lower sample size (n=9). All this information was included in the statistical analysis section of the paper. Moreover, the small sample size was included also as a limitation of the present work.
- Line 116: bopchemical, is this a typo?
Effectively. It has been modified.
- Lines 114-115 and 227-229 (discussion): Since the participants were healthy and of normal weight, can you clarify why plasma triglyceride levels were slightly higher after the 150 mg silibinin and thistle extract interventions compared to placebo? Additionally, could you provide the actual triglyceride values or ranges to contextualize this observation and confirm whether the increase is clinically meaningful?
This is an interesting comment. As also commented to reviewer 1, and consider the benefits of silibinin on liver conditions, the present data may indicate a higher clearance of liver triglycerides to the plasma. Nevertheless, the comment regarding actual triglycerides values is also of great importance, and it has been also included in the discussion.
- Page 4, Figure 1 and 2: This Figure only shows values of triglycerides not “different biochemical parameters measured”. Consider changing the captions to avoid confusion. This criterion should also be implemented for Figure 2 as this only shows waist circumference results.
Effectively, it was a mistake by our part. We have changes captions according to the reviewer’s comment.
- Please elaborate on the criteria used to determine the doses of silibinin used in this study.
In Spain, the Spanish Medicines Agency (AEMPS), have allowed the employment of silibinin at doses of 150 mg for therapeutic use (https://www.vademecum.es/espana/ficha-tecnica/2227/legalon-150-mg-capsulas). In addition, our previous in vitro results suggested that for a proper lipase inhibition, a dose near to 150 mg should be employed [5]. The highest dose was chosen considering that the present study was focused on studying the safety of the drug, also taking into account the data from the AEMPS, which confirmed that this drug had low toxicity. Regarding the milk thistle extract, it was selected because the commercial extracts available for use as medicines contain an approximate concentration of 150 mg of silibinin, which is precisely the concentration we wanted to test. All phases of the study were approved by the ethics committee of the Catholic University of Murcia (code: CE072111).
Results:
- I did not find any information on the results of the anthropometric parameters; including a table, either in the main text or as supplementary material, would be helpful. Same for body fat, waist circumference and other physical parameters measured in this study.
We regret this lack of information. We have included this information as supplementary table 1.
Discussion:
- Line 209: You mention that thistle extract had fewer adverse effects than orlistat. Could you clarify whether the adverse effects were dose-dependent or related to the specific nature of silibinin?
This is an interesting point, and although there will be a possible dose-dependent effect, in the present work, we observed more adverse effects with the lower silibinin dose. In fact, the higher silibinin dose was the treatment with the lower number of adverse effects. Overall, as all interventions showed lower adverse effects than orlistat, we can speculate that it was related to the specific nature of silibinin, rather than to the dose.
- Line 233-235: The authors mentioned reduction of abdominal fat but such information is not presented in the main text or supplementary information. Was abdominal fat actually measured? This is not similar to waist circumference and these results should not be overgeneralized.
Effectively, we regret this misunderstanding by our part. We have changed this paragraph in the revised version.
- Line 236: The authors mentioned several proposed mechanisms for silibinin's action on pancreatic lipase. Were any biomarkers or intermediate outcomes (e.g., postprandial lipid levels) to support these proposed mechanisms?
The comment of the reviewer is fairly interesting, but unfortunately, we do not evaluated these outcomes. In the previous report, we evaluated in deep the in vitro effect of silibinin, but in this time, we employed only fecal fat excretion as a measure of lipase inhibition, following the methodology of previous reports [4].
- Lines 236-251: Given that the study population consisted of healthy, normal-weight individuals, how generalizable are these findings to individuals with obesity, where pancreatic lipase activity and fat metabolism may differ?
There are two questions to consider. First, the benefits of silibinin on liver metabolism have been described both in normal weight and in people with overweight/obesity. Second, orlistat, other pancreatic lipase inhibitor, is effective in both weight ranges. Nevertheless, in the conclusion and in the limitation sections, we have clearly stated that the present work has been conducted in normal weight people, without evident metabolic complications, and therefore, further studies should be performed to confirm the suitability of silibinin as an effective agent for obesity treatment.
- Line 254: The authors proposed that other compounds in thistle extract may contribute to the effects observed. Please elaborate more on this to strengthen this discussion.
We fully agree with the reviewer. Therefore, we have extended this section of the discussion and included two new references to reinforce our hypotheses [6,7].
- Line 250-251: Please explain how these results can be translated to adults with obesity, given that this study was conducted in a different population. In my opinion, the results from this study are specific to healthy, normal-weight populations only. However, your findings provide a basis for methods that can be applied in future studies evaluating the effects of silibinin in populations with obesity, where physiological differences, such as altered lipid metabolism and microbiota composition, may yield distinct outcomes.
As commented previously, this is a limitation of the present study, and therefore, it has been included in this section. In addition, the conclusion of the study has also been modified.
- Lines 260 (gut microbiota results): Please clarify if you assess baseline microbiota composition in participants to evaluate whether individual starting profiles influenced their responses to silibinin. Also, how was the high inter-individual variability accounted for in microbiota composition during statistical analysis? Did you consider stratifying participants based on baseline microbiota profiles or other factors?
Effectively, baseline microbiota profiles were determined, and a new sentence has been included in the section 4.6 to clarify this issue. The inter-individual variability was evaluated according to the microbiome analyst procedure. A detailed description of the analysis is described in the same section. On the other hand, although it would be interesting to stratify the participants, as the reviewer fairly suggest, considering the low number of participants, such stratification may probably increase the probability of type II errors. Nevertheless, we consider inter-individual variability determining the alpha-diversity (described in supplementary figure S2).
- Line 266: Changes in Lachnospiraceae and SCFA production are mentioned. Can you provide evidence directly linking these changes to improved intestinal epithelial energy or lipid metabolism for silibinin?
In the present work we do not evaluated the SCFA production. Therefore, our observations were based on previous reports showing, on the one hand, that Lachnospiraceae are SCFA producing bacteria, and on the other hand, on the reports showing the benefits of SCFA on epithelial barrier and lipid metabolism (for instance, this issue has been reviewed in: Short chain fatty acids and its producing organisms: An overlooked therapy for IBD? Deleu, Sara et al. eBioMedicine, Volume 66, 103293)
- Lines 292-295: The genera Veillonellaceae and Flavobacteriaceae were positively correlated with fecal fat excretion. Could you elaborate on how these findings might influence silibinin’s efficacy in individuals with different microbiota profiles (e.g: obesity)?
This comment is also quite interesting. We have cited several references at this regard. In addition, we have included a new sentence to reinforce the relevance of this observation.
REFERENCES:
- Yi, M.; Manzoor, M.; Yang, M.; Zhang, H.; Wang, L.; Zhao, L.; Xiang, L.; Qi, J. Silymarin Targets the FXR Protein through Microbial Metabolite 7-Keto-Deoxycholic Acid to Treat MASLD in Obese Mice. Phytomedicine 2024, 133, doi:10.1016/J.PHYMED.2024.155947.
- Diab, F.; Zbeeb, H.; Zeaiter, L.; Baldini, F.; Pagano, A.; Minicozzi, V.; Vergani, L. Unraveling the Metabolic Activities of Bioactive Compounds on Cellular Models of Hepatosteatosis and Adipogenesis through Docking Analysis with PPARs. Sci Rep 2024, 14, doi:10.1038/S41598-024-78374-7.
- Yan, B.; Zheng, X.; Wang, Y.; Yang, J.; Zhu, X.; Qiu, M.; Xia, K.; Wang, Y.; Li, M.; Li, S.; et al. Liposome-Based Silibinin for Mitigating Nonalcoholic Fatty Liver Disease: Dual Effects via Parenteral and Intestinal Routes. ACS Pharmacol Transl Sci 2023, 6, 1909–1923, doi:10.1021/ACSPTSCI.3C00210.
- Bryson, A.; De La Motte, S.; Dunk, C. Reduction of Dietary Fat Absorption by the Novel Gastrointestinal Lipase Inhibitor Cetilistat in Healthy Volunteers. Br J Clin Pharmacol 2009, 67, 309–315, doi:10.1111/j.1365-2125.2008.03311.x.
- Del Castillo-Santaella, T.; Hernández-Morante, J.J.; Suárez-Olmos, J.; Maldonado-Valderrama, J.; Peña-García, J.; Martínez-Cortés, C.; Pérez-Sánchez, H. Identification of the Thistle Milk Component Silibinin ( A ) and Glutathione-Disulphide as Potential Inhibitors of the Pancreatic Lipase : Potential Implications on Weight Loss. 2021, 83, doi:10.1016/j.jff.2021.104479.
- Awla, N.J.; Naqishbandi, A.M.; Baqi, Y. Preventive and Therapeutic Effects of Silybum Marianum Seed Extract Rich in Silydianin and Silychristin in a Rat Model of Metabolic Syndrome. ACS Pharmacol Transl Sci 2023, 6, 1715–1723, doi:10.1021/acsptsci.3c00171.
- Chen, W.; Zhao, X.; Huang, Z.; Luo, S.; Zhang, X.; Sun, W.; Lan, T.; He, R. Determination of Flavonolignan Compositional Ratios in Silybum Marianum (Milk Thistle) Extracts Using High-Performance Liquid Chromatography. Molecules 2024, 29, doi:10.3390/MOLECULES29132949.

Round 2
Reviewer 1 Report
Comments and Suggestions for Authors
Manuscript titled “Effect of silibinin on human pancreatic lipase inhibition and gut microbiota in healthy volunteers: a randomized controlled trial” reports a study on human participants, who were treated with either orlistat, a thistle extract or silibinin, in order to analyze the effects on weight loss and intestinal microbiota of the latter two treatments. The present version of the document was modified, according to comments and suggestions made during an initial revision, those made by the present reviewer include:
1. Fixing the colors of Figures 1 and 2, in order to see the individual data points. The colors of the figures were corrected.
2. Maintaining the same colors for each treatment on all figures. The colors have been homogenized.
3. Clarifying how the participant’s hunger/satiety was analyzed. The authors have now specified how this was determined.
4. Adding the units to the value shown for orlistat (15.8 ± 25.2), and confirming if the error is correct. The authors added the corresponding units to this value and corrected the error.
5. Defining the meaning of the asterisks in Table 3. The authors confirmed that they were an error, and have now removed them.
6. Rephrasing a sentence that highlights the suitability of long-term use of the treatments in obese individuals. The phrase was rewritten to more appropriately reflect the data in the present work.
7. Restructuring a sentence regarding changes in lipids and hepatoprotection. The phrase was modified.
8. Including a limitation regarding the participation of normo-weight individuals. The limitation was included.
9. Rephrasing the conclusion, in order to better reflect the data reported in the study. The phrase was modified.
According to the aforementioned changes made by the authors, it appears that they adequately considered the comments and suggestions made by the present reviewer. There are no additional comments for the revised version of the document.